# Optimized methods for scRNA-seq and snRNA-seq of skeletal muscle stored in nucleic acid stabilizing preservative
Elisabeth F. Heuston [1] ✉, Ayo P. Doumatey [1], Faiza Naz[2], Shamima Islam[2], Stacie Anderson[3], Martha R. Kirby[3], Stephen Wincovitch[4], Stefania Dell'Orso[2], Charles N. Rotimi [1] & Adebowale A. Adeyemo [1] ✉

Single cell studies have transformed our understanding of cellular heterogeneity in disease but the need for fresh starting material can be an obstacle, especially in the context of international multicenter studies and archived tissue. We developed a protocol to obtain high-quality cells and nuclei from dissected human skeletal muscle archived in the preservative Allprotect® Tissue Reagent. After fluorescent imaging microscopy confirmed intact nuclei, we performed four protocol variations that compared sequencing metrics between cells and nuclei enriched by either filtering or flow cytometry sorting. Cells and nuclei (either sorted or filtered) produced statistically identical transcriptional profiles and recapitulated 8 cell types present in skeletal muscle. Flow cytometry sorting successfully enriched for higher-quality cells and nuclei but resulted in an overall decrease in input material. Our protocol provides an important resource for obtaining high-quality single cell genomic material from archived tissue and to streamline global collaborative efforts.

Fluid-based, gravity-based, and other methods of single-cell sequencing provide a fine-grained understanding of cellular heterogeneity in health and disease that maximizes the amount of information that can be obtained from limited quantities of tissue. Although freshly isolated tissue gives the greatest likelihood of generating high-quality single-cell data, for many studies, this is not available and not practical in several under-resourced research settings globally, with the potential of exacerbating single-cell research inequity. Thus, significant effort has been spent on extracting high-quality data from fixed or banked tissue[1–3] using methods that include methanol fixation, dimethyl sulfoxide (DMSO) cryopreservation, stabilization in Lomant's reagent, CellCover reagent, and cross-link reversal of paraformaldehyde-fixed tissues[1–4]. Depending on the tissue type, origin, and storage conditions, different approaches offer different sets of advantages for obtaining the optimal amount of high-quality data.

Multicenter research studies often collect tissue from multiple field sites, necessitating careful preservation of the biospecimens from collection, through transportation, and ultimately the laboratory responsible for downstream processing. This issue has led to the development of a variety of tissue preservation methods. For genomics and transcriptomics applications, some of the most popular methods include Allprotect® Tissue

Reagent (ATR) and RNAlater®. These and other stabilization agents also offer the advantage of being less temperature sensitive[5–8], allowing specimens to be collected at a wider variety of field sites. However, many of these methods were developed in the context of bulk tissue assays and have not been evaluated outside this context. Of the most popular methods, the stabilizing reagent ATR offers the most promise for adaptation to field studies and multicenter collaborative efforts because it allows biospecimens to be stored at 37 °C for up to 24 h, and archived once tissues are transferred to lower temperatures[9]. ATR has been tested for transcriptional profiling by bulk RNA sequencing and for other biomolecules[6,10], but its application for single-cell technology is untested.

In the present study, we investigated whether tissue stored in ATR was suitable for single-cell transcription profiling. Using ATR-stored *vastus lateralis* (skeletal muscle) tissue from Nigerian participants enrolled in the Africa America Diabetes Mellitus (AADM)[11], we developed and compared four processing pipelines. Given arguments that single nucleus RNA-Seq (snRNA-Seq) is more powerful at assaying solid tissues than single-cell RNA-Seq (scRNA-Seq)[12–14], we compared whole cell versus nuclei preparations. To control for cellular lysis and debris common when working with tissues, we tested the effect of using flow cytometry sorting (fluorescence-

[1]Center for Research on Genomics and Global Health, National Human Genome Research Institute, National Institutes of Health, Bethesda, MD, 20892, USA. [2]Genomic Technology Section, National Institute of Arthritis and Musculoskeletal and Skin Diseases, National Institutes of Health, Bethesda, MD, USA. [3]NHGRI Flow Cytometry Core, National Human Genome Research Institute, National Institutes of Health, Bethesda, MD, 20892, USA. [4]Advanced Imaging & Analysis Core, National Human Genome Research Institute, NIH, Bethesda, MD, USA. ✉e-mail: heustonef@nih.gov; adeyemoa@mail.nih.gov

activated cell sorting; FACS) to enrich intact nuclei by staining for nuclear pore complex proteins. All four methods yielded data with expected skeletal muscle transcriptional profiles, with similar sequence quality RNA profiling results between cell and nuclei preparations. Notably, FACS enrichment significantly depleted low-quality input material before sequencing, but had a negative impact on sequencing reaction yield. Our pipeline demonstrates that stabilization of skeletal muscle tissue in ATR is of sufficient quality for single-cell transcriptional assays and provides a means of expanding high-throughput genomics through diverse populations across the globe.

## Results

### Tissue dissociation and sample capture

The workflow of our methods for tissue dissociation, sample capture, and downstream analyses are summarized in Fig. 1 and detailed in Supplemental Methods. Briefly, we washed and dissociated stored tissue before staining with antibodies detecting nuclear pore complex proteins (NPC) to determine if membranes in ATR-stabilized cells were intact. NPC+ cells were isolated using fluorescence-activated cell sorting (FACS) and subsequently stained with DAPI. Fluorescent imaging showed intact nuclei present within the stored tissue, suggesting the nuclear-based single-cell genomic assays were feasible (Fig. 2a).

Since the quantity and quality of banked tissue varies greatly, we tested four tissue protocols prior to 10X genomics gel beads in-emulsion (GEM) capture (Fig. 1b). First, we tested whether sample (where "sample" collectively refers to both cells and nuclei) impacted the number of genes detected in downstream analysis. Second, we tested whether the preparation method (collectively referring to performing sample filtration alone or sample filtration followed by FACS enrichment, see Methods) improved data quality. To demonstrate protocol efficacy, two biospecimens were selected from

apparently healthy individuals and pooled to create 142 mg of total tissue (Fig. 1b). Approximately $7.4 \times 10^5$ cells were recovered after homogenization and sequential filtration steps. Samples (pooled cells or nuclei) were divided for one of four protocols and loaded according to manufacturer guidelines: whole cell or nuclei samples, with or without sorting preparations. Note that when working with nuclei from a sorted preparation, we are referring to cells that were sorted prior to nuclei extraction. A summary of capture statistics is shown in Table 1.

### Data quality and clustering

All samples from each preparation were sequenced using 5′ gene expression chemistry (10x Genomics). Demultiplexed data from each of the four protocols were assessed by multiple quality control metrics and filtered according to gene counts and percentage reads mapping to mitochondrial RNAs (PMMR) (Fig. 1c and Supplementary Fig. 1). Simulated doublet detection demonstrated that 95 to 99% of all samples were singlets, suggesting that protocol differences did not significantly impact doublet rate (Table 1). Ultimately, sequencing data for 40,612 samples (cells and nuclei) remained for downstream analysis. Correlating UMI and gene counts showed an overall similarity between the four protocols ($\rho = 0.94$), with individual correlations per protocol ranging between $\rho = 0.88$ and $\rho = 0.96$ (Supplementary Fig. 2). We observed differences between protocols regarding the distribution of UMI counts across clusters, genes detected per sample, and PMMR ranges, which are explored in more detail below (Fig. 2b, c and Supplementary Fig. 2).

Because all samples originated from the same tissue pool and were sequenced in a single batch, we can attribute variations in RNA levels to differences between the four protocols. Out of 24,909 detected genes, 44 had a standardized variance >3, while an additional 59 had a standardized

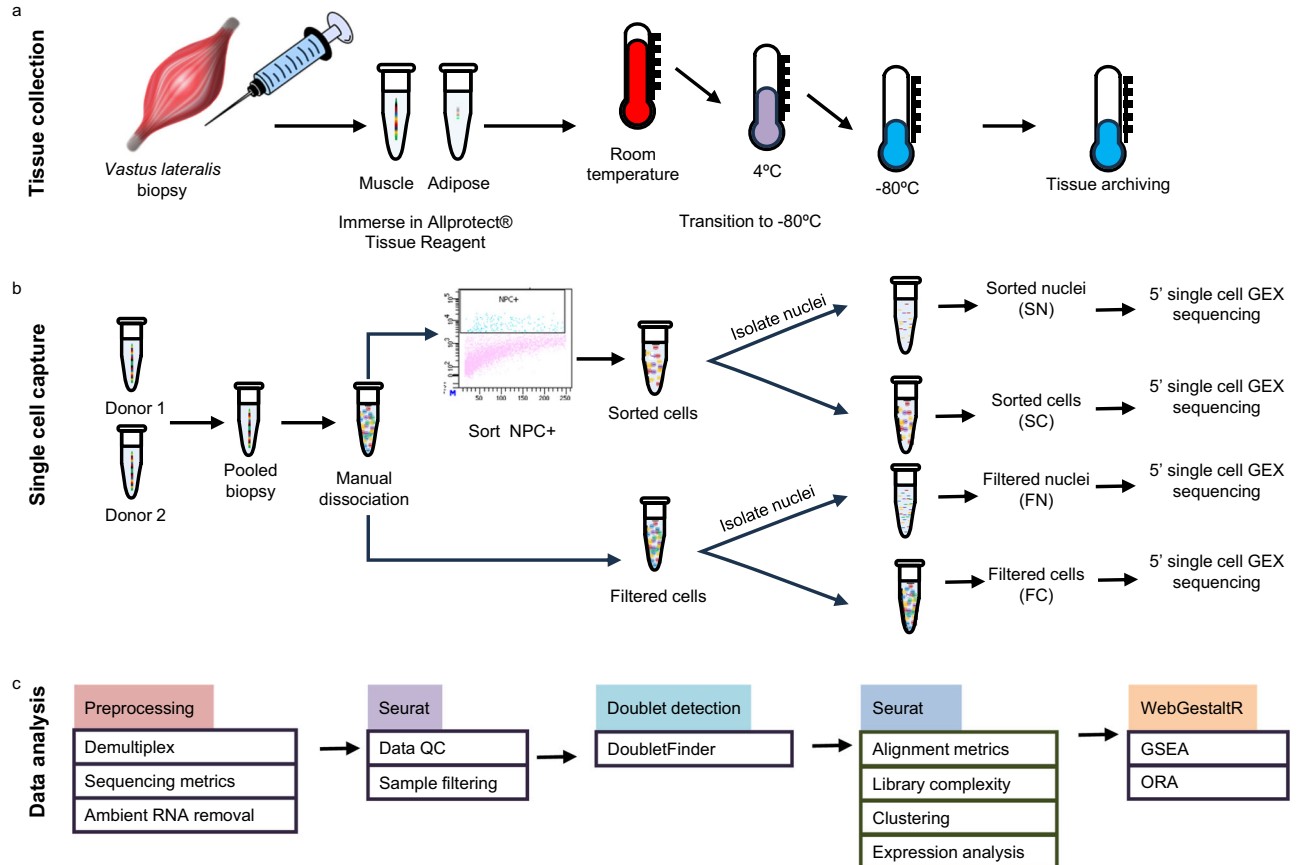

**Fig. 1 | Experimental design. a** Needle biopsy tissue was collected from adult individuals enrolled in the AADM study, placed in ATR, and transitioned to longer-term storage temperatures before shipment to a central processing center. **b** Single-

cell capture protocol. Samples from two donors were pooled before manual dissociation and subsequent processing. **c** Data analysis pipeline as detailed in Methods.

**Fig. 2 | Population characteristics of cells and nuclei. a** Confocal imaging of cells (top) and nuclei (bottom) stained with DAPI and NPC-AF594. **b** Violin plot indicating the number of genes detected per sample after filtering. **c** Violin plot indicating the percentage of reads mapping to mitochondrial RNAs in each sample after filtering. **d** Over representation analysis of 103 genes with the highest standardized variance. **e** UMAP of pooled samples, colored by cluster assignment. * Box plots show median, 25th, and 75th percentiles. * FC filtered cells, FN filtered nuclei, SC sorted cells, SN sorted nuclei.

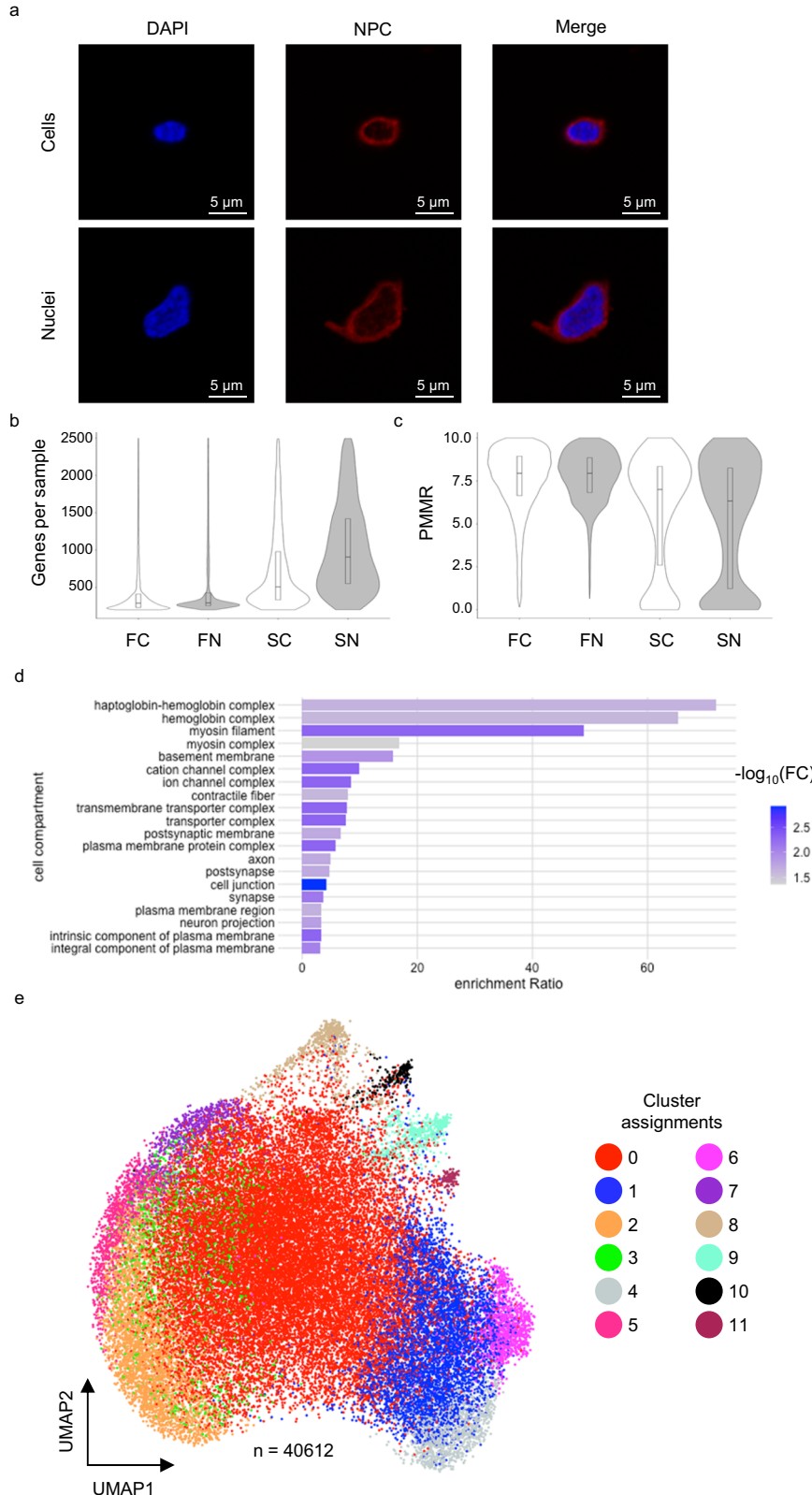

variance >2.5 (Supplementary Data 1). Erythrocyte genes *HBB*, *HBA2*, and *HBA1* were among the top ten with the highest standardized variance, as well as *DLC1*, *LRRTM4*, *PDZRN4*, *MEG3*, and *CACNA1C*. Using over representation analysis (ORA), the 103 highly variable genes localized to cell compartments associated with hemoglobin biology (FDR <0.03), myosin filaments (FDR <0.005), basement membrane (FDR = 0.01), and cell junctions (FDR = 0.02) (Fig. 2d). These results suggest that the largest sources of variance among the protocols include erythrocyte populations and cellular debris but is unlikely to impact insights into skeletal muscle biology.

## Skeletal muscle cell populations are recapitulated in transcriptional data

We pooled transcriptional data from each of the four protocols (cells, nuclei, filtered, sorted) for Louvain clustering. Applying a dimensional reduction

threshold that accounts for 80% of gene expression variance generated 12 transcriptionally distinct clusters (Fig. 2e and Table 2). We then performed differential expression analysis to identify markers of each cluster, defining significance as FDR ≤ 0.05, log2FC ≥ |1.5| (Supplementary Data 2). Within these parameters, all 12 clusters could be described by a unique set of ≥10 differentially expressed transcripts.

To determine if the RNA profiles identified in these protocols recapitulated the expected cell types present in skeletal muscle, clusters were annotated using gene profiles collected from multiple sources[15–17]. We identified 5 distinct skeletal muscle clusters expressing canonical markers, including those of type I (*MYH7*), type IIA (*MYH2*), or type IIX (*MYH8*) fibers (Fig. 3 and Table 2). Endothelial (*VWF, PECAM1*), smooth muscle (*PLN*), fibroblast and fibroblast/adipocyte progenitor (*COL3A1, DCN, FBN1*), pericyte (*NOTCH3, RGS5*), and satellite (*APOE, PAX7*) cell clusters were also present. These data show that biospecimens stored in ATR retain the variety of skeletal muscle cell types present in vivo.

When characterizing clusters by significant gene expression, cells in clusters 0 and 3 were only defined by genes that were expressed lower (log2FC ≤ −1.5, FDR ≤ 0.05) relative to other clusters (Supplementary Data 2). These genes included *MUC3A, NFASC, NBPF15* (cluster 0), *SUFU, NPIPB4*, and *MROH7* (cluster 3). To assign putative identities to these clusters, we performed ORA using the top 50 genes with the highest log2FC >0 (including *HBB, HIST1H4C*, and *RPL38* from cluster 0 and *RPS20,*

*RPS16*, and *UQCR10* from cluster 3) and found significant (FDR ≤0.05) enrichment of mitochondrial-related pathways, protein transport processes, and RNA processing (Supplementary Fig. 3). Compared to the other 10 clusters, clusters 0 and 3 had the lowest median gene and UMI counts (Supplementary Fig. 3). Finally, no simulated doublets were assigned to these clusters during doublet scoring. Based on this information, we conclude that clusters 0 and 3 represent erythrocytes and cellular debris.

## Comparison of cell and nuclei sampling

Given that ATR-preserved cells retained the expected variety of cell types present in skeletal muscle, we investigated whether protocols altered the detected transcriptional profile. After quality control, 19,986 cells and 20626 nuclei remained for downstream analysis (Fig. 4a and Table 1). No significant differences were observed in data quality: the median (IQR) number of genes per sample was 301 (235–456) in cells and 301 (258–636) in nuclei, and the PMMR was 7.87 (6.49–8.89) in cells and 7.87 (6.66–8.82) in nuclei (Fig. 4b, c). We compared gene expression profiles between filtered and sorted populations to determine whether sample type influences which cellular processes are detected. Differential gene expression revealed only five genes at an FDR ≤0.05 and log2(FC) ≥|1.5|: *DSCAM, ABCB1, RASGEF1B, SLC26A3, CSMD1* (Supplementary Data 3). Most of these genes are involved in transmembrane activities and likely reflect the different RNA species that exist within the cytoplasm and nuclear compartments. Finally, if the sample type does not influence measured gene expression, both cells and nuclei should have equal percentages of samples assigned to each cluster. We used a neighborhood-based approach[18] to test for differential abundance between cells and nuclei across the 12 clusters. Of 1392 defined neighborhoods, no significant enrichment (adjusted FDR ≤0.05) was found in any cluster, demonstrating that assayed cells and nuclei represent the same transcriptional profiles (Fig. 4d and Table 2).

## Comparison of sorted and filtered methods

We next tested whether sorting to remove debris affected the transcriptional profiles that were detected. Despite loading similar numbers of samples for filtered and sorted runs, ~53% (*n* = 38,782) of filtered samples were captured, compared to only ~4% (*n* = 3967) of sorted samples (Fig. 4e and Table 1). The two preparations also had a number of differences regarding data metrics. The median (IQR) number of genes per sample was 291 (245–417)

## Table 1 | Summary of capture statistics for each protocol

| Sample | Preparation | Captured | No. samples after gene level-based filtration* (% of captured) | No. samples after simulated doublet detection** (% of UMI-filtered) |
|---|---|---|---|---|
| Cells | Filtered | 48,802 | 18,805 (39%) | 17,958 (95%) |
| Nuclei | Filtered | 24,598 | 19,977 (81%) | 18,704 (94%) |
| Cells | Sorted | 3470 | 2031 (59%) | 2028 (99%) |
| Nuclei | Sorted | 3995 | 1936 (48%) | 1922 (99%) |

* UMI-filtration excluded cells with ≤200 genes, ≥2500 genes, or >10% reads mapped to mitochondrial RNAs.

** Simulated doublet detection and exclusion was performed with DoubletFinder[24].

## Table 2 | Cluster identities and characteristics

| Cluster | Cell count | Sample | | Method | | Top three positively expressed genes* | Putative identity |
|---|---|---|---|---|---|---|---|
| | | Cells | Nuclei | Filtered | Sorted | | |
| 0 | 23307 | 51.6% | 63.0% | 63.0% | 5.5% | *HIST1H4C, MIR4458HG, HBB* | Erythrocytes/debris |
| 1 | 5586 | 18.0% | 9.7% | 12.6% | 25.0% | *ATG16L1, ITIH6, OSBP2* | Myocyte [skeletal muscle] |
| 2 | 4044 | 10.9% | 9.0% | 8.5% | 23.6% | *ALDOA, TEX36, WNT4* | Myocyte [skeletal muscle] |
| 3 | 1634 | 2.3% | 5.7% | 4.4% | 0.2% | *RPS16, UBAS2, RPL37a* | Erythrocytes/debris |
| 4 | 1166 | 3.4% | 2.4% | 1.5% | 15.7% | *SLA2, LRRC37A, SPDYE16* | Myocyte [skeletal muscle] |
| 5 | 1164 | 2.1% | 3.6% | 2.7% | 4.0% | *PRRT4, PARD6A, PYCR3* | Myocyte [skeletal muscle] |
| 6 | 1150 | 3.6% | 2.1% | 1.8% | 12.2% | *CYP4F26P, SLC5A11, CD24* | Myocyte [skeletal muscle] |
| 7 | 798 | 2.6% | 1.4% | 1.7% | 4.1% | *SMARCA5-AS1, DDN, PARD6A* | Myocyte [smooth muscle] |
| 8 | 708 | 2.2% | 1.3% | 1.6% | 3.4% | *TCTEX1D1, HCRTR1, P2RY8* | Endothelial cells |
| 9 | 625 | 2.0% | 1.1% | 1.3% | 3.7% | *KRT19, ELANE, CDH10* | Fibroblast/adipocyte progenitors |
| 10 | 253 | 0.8% | 0.4% | 0.5% | 1.8% | *NPYSR, GPR63, OR7C1* | Pericytes |
| 11 | 177 | 0.6% | 0.3% | 0.4% | 0.9% | *APOC2, KCNK17, CECAM16* | Satellite cells |

* Restricted to RefSeq genes with validated status.

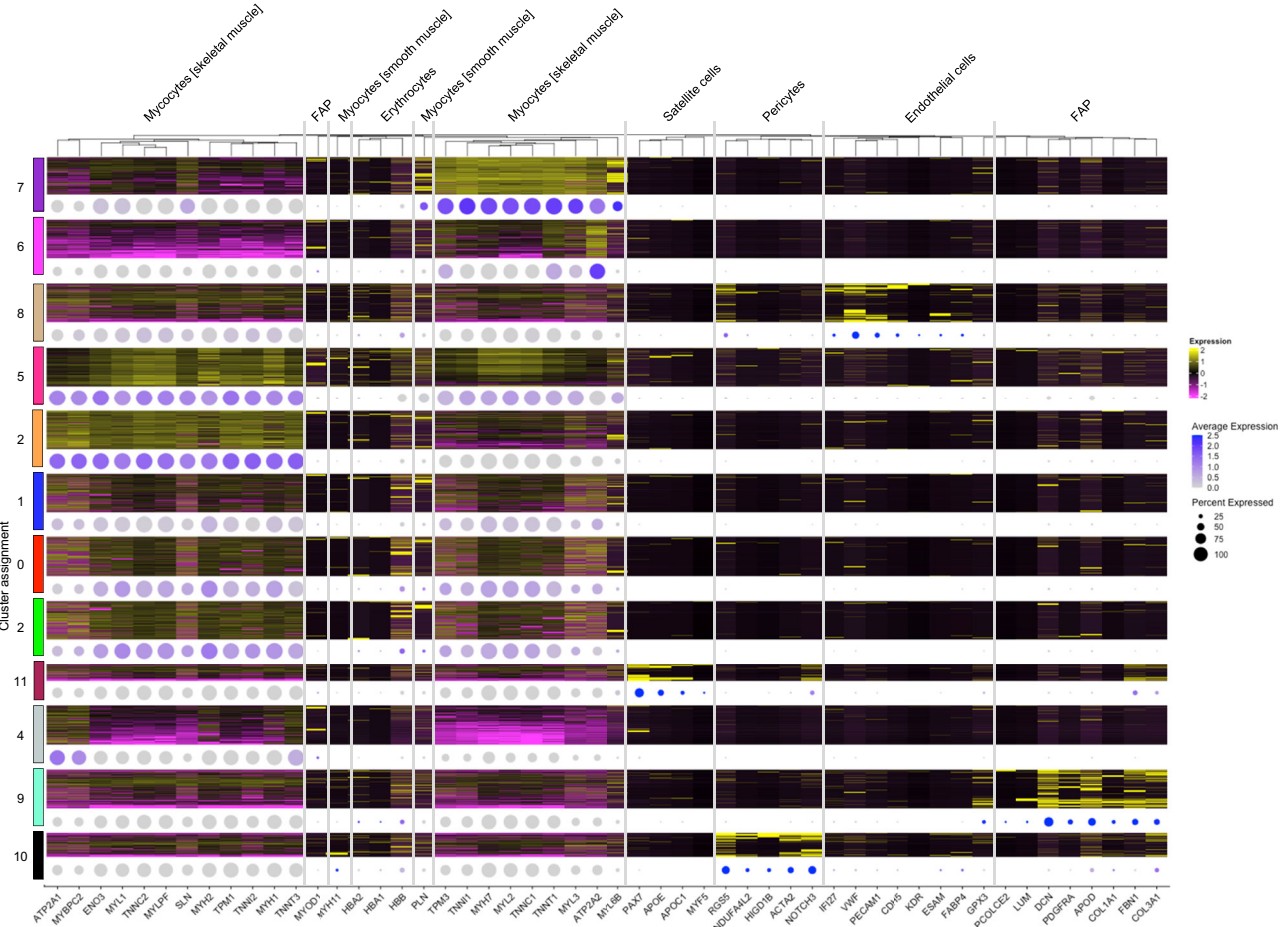

**Fig. 3 | Profiles of marker genes expressed in each cluster.** Columns show unsupervised hierarchical clustering of scaled gene expression; rows represent cells grouped by cluster. Heatmap shows relative expression per cell of marker genes (yellow = high, magenta = low). Dot plot of average gene expression per cluster, with color indicating average expression (purple = high, gray = low) and dot size indicating the percentage of cells in the cluster expressing the gene. Heatmap clusters are randomly downsampled to ≤400 cells. FAP fibroblast/adipocyte progenitor cells.

in filtered preparations, compared to 702 (390–1242) in sorted preparations (Fig. 4f). In contrast, filtered preparations had higher PMMR (7.94 (6.74–8.89)) compared to sorted preparations (6.74 (1.83–8.31), Fig. 4g). Differential gene expression analysis to identify preparation-specific RNA profiles revealed 357 genes differentially expressed at FDR ≤0.5 and log2(FC) ≥[1.5] (Supplementary Data 4). Gene set enrichment analysis (GSEA) showed enrichment of genes located in the mitochondrial inner membrane (FDR = 0.001, NES = 2.2) and electron transport chain (FDR = 2.2E-16, NES = 3.0), suggesting that the predominant difference is the presence of mitochondrial RNAs in filtered samples (Supplementary Fig. 4d).

Clustering analysis assigned 67% of all filtered samples to clusters 0 and 3 (identified as erythrocytes and cellular debris based on gene expression profiles), compared to only 5.5% of sorted samples (Table 2). We performed differential abundance analysis to confirm the biased assignment of filtered samples in clusters 0 and 3 and found significant neighborhood enrichment (adjusted FDR ≤0.05) of filtered samples in clusters 0 ($n = 645$) and 3 ($n = 48$) (Fig. 4h). Removing clusters 0 and 3 recapitulated the UMAP profiles of sorted samples (Fig. 4i). The median (IQR) profiles for the number of genes detected per sample (554 (323 – 1054) and 712 (391–1258) for filtered and sorted, respectively) and percentage of reads mapping to mitochondrial RNAs (6.84 (5.47–8.32) and 6.54 (1.4–8.21) for filtered and sorted, respectively) also became more similar (Fig. 4j, k). Finally, differential abundance testing after the removal of clusters 0 and 3 identified minimal differences between the populations (Fig. 4l). Based on these data, we conclude that clusters 0 and 3 contain erythrocyte precursors and/or cell debris that were removed by sorting.

## Discussion

We showed that archived skeletal muscle stored in Allprotect® Tissue Reagent (ATR) produces high-quality data suitable for single-cell and single-nucleus RNA sequencing. Skeletal muscle is comprised of cells from many populations, including progenitor, muscle, epidermal, adipose, and blood cells. We developed four protocols to test sequence read quality using: filtered cells, filtered nuclei, sorted cells, and sorted nuclei. Using 5' gene expression profiling, we demonstrated that all four of these methods generate biologically relevant profiles that encompass the heterogeneity of skeletal muscle.

Noting reported biases using single-cell versus single-nucleus RNA sequencing on solid tissue, we performed a head-to-head comparison of these methods. A droplet-based platform, like the Chromium system from 10X Genomics, offers both high-throughput and high-sensitivity sequencing, thereby increasing the likelihood of achieving quality sequence data by sheer number of processed units. Additionally, 5' RNA sequencing produces information on transcriptional start site (TSS) usage, allowing better profiling of the genes that define each population and helping to overcome differences between detected RNAs in nuclear versus cytoplasmic and nuclear transcriptomes. Finally, while cell versus nuclei biases have been reported in freshly isolated tissue, it is important to account for the different conditions present when using archived tissue, for which droplet-based systems perform well in benchmark tests[19,20]. These factors help explain the similarities we observe between these procedures.

Another significant consideration when designing single-cell or single-nucleus genomic experiments is the enrichment of the population of

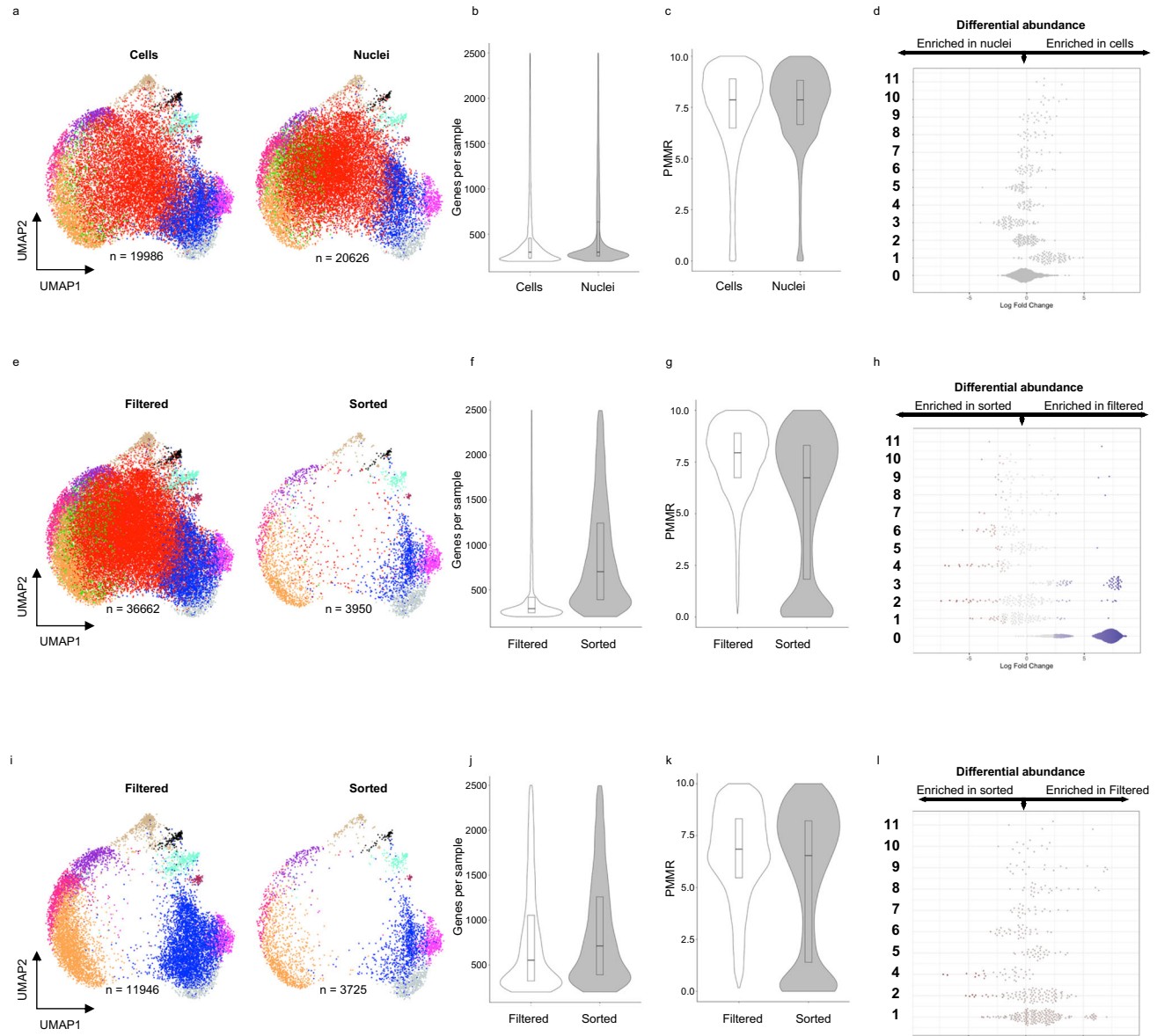

**Fig. 4 | Data metrics and clustering comparing four protocols. a** UMAP of pooled cell (left) or nuclei (right) samples. **b** Violin plot indicating the number of genes detected per sample after filtering. **c** Violin plot indicating the percentage of reads mapping to mitochondrial RNAs per sample after filtering. **d** Differential abundance testing using neighborhood analysis. **e** UMAP of pooled filtered (left) or sorted (right) preparations. **f** Violin plot indicating the number of genes detected per sample after filtering. **g** Violin plot indicating the percentage of reads mapping to mitochondrial RNAs per sample after filtering. **h** Differential abundance testing using neighborhoods. **i** UMAP of pooled filtered (left) or sorted (right) preparations after removal of clusters 0 and 3. **j** Violin plot indicating the number of genes detected per sample after filtering after removal of clusters 0 and 3. **k** Violin plot indicating the percentage of reads mapping to mitochondrial RNAs per sample after filtering after removal of clusters 0 and 3. **l** Differential abundance testing using neighborhoods. Clusters are indicated in bold. Italics indicate the percentage of all nuclei (left) or all cells (right) assigned to the cluster after removal of clusters 0 and 3. Bar plots indicate a median with 25 to 75% percentiles marked.

interest. Filtering samples with sequential 70 and 40 μm strainers is a standard step in "single unit" genomic assays, but this is an agnostic method using size-based exclusion. Magnetic bead and fluorescence-activated surface-marker approaches allow more targeted enrichment at the expense of additional sample handling and potential loss of novel populations that don't express the marker of interest. By comparing filtering alone against sorting using an antibody against nuclear pore complex proteins, our goal was to maintain agnostic enrichment of high-quality cells and nuclei while excluding sample debris. Our results showed that the total number of captured samples (cells or nuclei) was higher without sorting, but a much larger proportion of the captured samples were identified as debris and red blood cells. Thus, when the amount of tissue is not a limiting factor, sorting may increase the capture efficiency of high-quality samples and increase the

chances of identifying low-frequency populations. However, when the amount of tissue is limited, it may be advantageous to avoid targeted enrichment with the goal of maximizing the number of captured samples.

While our protocol demonstrates a method of single-cell RNA sequencing from archived tissues, there are a few limitations to note. Firstly, our method has been optimized specifically for human skeletal muscle, and analysis of other human tissues may require optimization. Additionally, ATR is known to cause cell shrinkage, rendering it inappropriate for morphological studies. Finally, ATR can denature proteins, and thus, caution should be used with selecting antibodies with conformational epitopes or when assaying for intact chromatin.

In summary, we have shown that we can generate high-quality scRNA-seq and snRNA-seq data from skeletal muscle preserved in a nucleic acid

preservative agent (ATR). While fresh tissue is still the sequencing gold standard for both quality and quantity, stabilizing tissue in ATR facilitates studies that collect biospecimens from multiple study sites, as individual sites do not need access to equivalent processing equipment and expertise. Biospecimens can be shipped and processed at a single location or a subset of locations to mitigate technical variation and batch effects. The demonstration that skeletal muscle tissue preserved in ATR (a commonly used stabilizing reagent) yields high-quality sequencing material from both cells and nuclei also offers the opportunity to study archived clinical and research tissue biospecimens. Our protocol presents researchers with opportunities to expand the applicability of high-throughput single-sample sequencing to a greater diversity of research and clinical contexts.

## Methods

### Tissue collection
Tissue specimens were selected from adult Nigerians enrolled in the Africa America Diabetes Mellitus (AADM) study, which is a large and long-standing genetic epidemiology study of type 2 diabetes (T2D) in sub-Saharan Africa[11,21]. T2D was defined using the American Diabetes Association (ADA) criteria. The study protocol was approved by the institutional ethics review board (IRB) of the National Institutes of Health/National Human Genome Research Institute (protocol HG-09-N070) and the IRBs of each participating institution, including University of Lagos, Lagos, Nigeria; College of Medicine, University of Ibadan, Ibadan, Nigeria; University of Nigeria Teaching Hospital, Enugu, Nigeria; University of Science and Technology, Kumasi, Ghana; University of Ghana Medical School, Accra, Ghana. Written informed consent was obtained from each participant prior to enrollment.

For a subset of participants, skeletal muscle samples were obtained from the vastus lateralis muscle under local anesthesia using the VacoraTM Vacuum-Assisted Biopsy System, a handheld biopsy system capable of collecting up to 170 mg of tissue in one pass. The vastus lateralis muscle was used as the biopsy site because of its bulk, distance from major vascular structures, and easy accessibility. The resulting samples were quickly dissected into visible skeletal muscle and adipose tissue components, and each component was immediately placed in a 15-ml prelabelled Falcon centrifuge tube and prefilled with at least 5 ml of Allprotect® Tissue Reagent (ATR) (Qiagen, Germantown, MD, catalog no. 76405) to stabilize tissue RNA, DNA and proteins. The AllProtect volume was accordingly adjusted to the size of the tissue aliquot which in our study varies between 50 mg to 180 mg to ensure complete submersion of the tissue samples in the collection tubes. The collection tubes were then placed in an ice bucket and transferred from the clinic to the lab in a couple of hours. For archival of the tissue samples at −80 °C, they are kept overnight in the solution at 4–8 °C and then transferred to −80 °C until shipped to the coordinating center, the Center for Research on Genomics and Global Health at the National Institutes of Health (Bethesda, MD, USA) from the study site (Ibadan, Nigeria). Shipping of the samples was done on dry ice.

### Tissue processing
A detailed protocol of cell isolation and imaging is described in Supplemental Note 1 and is available at www.protocols.io. Briefly, skeletal muscle specimens stored in Allprotect® Tissue Reagent (ATR) were thawed on ice. Ice-cold PBS was added and pipette-mixed with ATR to dilute and remove supernatant. This process was repeated until all ATR was removed. Tissue was subsequently rinsed twice in ice-cold PBS to remove any remaining ATR, weighed, and resuspended in pBSA (PBS + 0.04% BSA) + 1 U/mL RNaseOut (RNaseOUT™ Recombinant RNase Inhibitor, Thermo Fisher Scientific, Waltham, MA, catalog no. 10777019). Manual tissue dissociation was performed with a disposable micro-tissue homogenizer. Cell concentrations were estimated with a hemocytometer. Single-cell suspensions were strained sequentially through 70 and 40 µm filters, with washes between each step ("filtration alone" samples).

For sorted samples, cells were stained with a final concentration of 25 µg/mL anti-Nuclear Pore Complex conjugated to Alexa Fluor® 594

(NPC-AF594) antibody (BioLegend, San Diego, CA, catalog no. 682202) at 4 °C, 30 min. Excess antibody was removed and NPC+ cells isolated on a Becton Dickinson FACS ARIA Fusion using a 70 µm nozzle.

For nuclei isolation, cells were incubated with a Nuclear Extraction Buffer (Miltenyi Biotec, Gaithersburg, MD, catalog no. 130-128-024) containing 1 U/mL RNaseOut at 4 °C for 5 min. Following the incubation, suspensions were centrifuged at 300×$g$, 4 °C, 5 min, supernatant removed, and nuclei resuspended in pBSA. Samples were sorted on ice into pBSA before downstream processing. Note that "sorted nuclei" refers to cells that were sorted prior to nuclei extraction.

### Cell and nuclear imaging
Sorted cells were incubated with an additional 100 µg/mL NPC-AF594 antibody + 5 µg per DAPI per 100 µL of sample DAPI for 20 min at 4 °C. Cells were washed and resuspended in 200 µL and plated in a 35 mm 1.5 coverslip dish (MatTek). Imaging was done on an LSM 880 Airyscan Confocal microscope (Zeiss) with at 63x/1.4 oil objective. Images were captured with Zen_2.3 software (Zeiss).

### 10X Genomics capture
Cells were loaded on a Chromium Instrument (10x Genomics, Pleasanton, CA) to generate single-cell GEMs. Single-cell RNA-seq libraries were prepared using a Chromium Single Cell 5' Library & Gel Bead Kit v2 (P/N 1000263, 10x Genomics). GEM-RT was performed in a C1000 Touch Thermal cycler with 96-Deep Well Reaction Module (Bio-Rad; P/N 1851197): 53 °C for 45 min, 85 °C for 5 min; held at 4 °C. Following retro-transcription, GEMs were broken, and the single-strand cDNA was purified with DynaBeads MyOne Silane Beads. cDNA was amplified using the C1000 Touch Thermal cycler with 96-Deep Well Reaction Module: 98 °C for 3 min; cycled 12 times: 98 °C for 15 s, 63 °C for 20 s, and 72 °C for 1 min; 72 °C for 1 min; held at 4 °C. Amplified cDNA product was purified with the SPRIselect Reagent Kit (0.6× SPRI). Indexed sequencing libraries were constructed using the reagents in the Chromium Single Library construction kit, following these steps: (1) end repair and A-tailing; (2) adapter ligation; (3) post-fragmentation, end repair, and A-tailing double size selection cleanup with SPRI-select beads; (4) sample index PCR and cleanup. The barcoded sequencing libraries were diluted at 3 nM and sequenced on Illumina NovaSeqX or Illumina NexSeq2000 using the following read length: 28 bp for Read1, 10 bp for I7 Index, 10 bp for I5 Index and 90 bp for Read2.

### Data processing
FASTQ files were demultiplexed in Cell Ranger_7.2.0 (10x Genomics) and mapped to the GRCh38-2020-A transcriptome. Ambient RNA excluded with CellBender_0.3.0[22]. Quality control and clustering was performed with Seurat_5.0.1[23], SeuratObject_5.0.0, and DoubletFinder_2.0.3[24]. Filtering was performed as follows: nFeature_RNA ≥ 200, nFeature_RNA ≤ 2500, percentage reads mapped mitochondrial RNAs (PMMR) ≤10. The predicted doublet rate was set at 0.05. Because all specimens were derived from the same starting material and sequenced in a single batch, data were log normalized with a scale factor of 10000. Differential abundance testing was performed with MiloR_1.8.1[18]. Over representation analyses (ORA) and gene set enrichment analyses (GSEA) were performed in WebGestalt_0.4.6[25].

### Statistics and reproducibility
Statistical tests used to identify the significance levels have been described in "Results" and "Methods", and in figure legends. To generate single cell RNA-Seq data, skeletal muscle from three independent donors were homogenized and pooled before subsequent processing. Source data and code availability are described in "Methods".

### Reporting summary
Further information on research design is available in the Nature Portfolio Reporting Summary linked to this article.

## Data availability

All data relevant to this study are available through the Gene Expression Omnibus at GSE282362. Source data can be obtained in Supplementary Data 1–4.

## Code availability

The code necessary to reproduce the primary results of this study are available at the git repository https://github.com/heustonefNIH/skeletal_muscle.

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

## Acknowledgements

The contents of this publication are solely the responsibility of the authors and do not necessarily represent the official view of the National Institutes of Health. The authors gratefully acknowledge the AADM participants, their physicians, AADM site investigators - Drs. Clement A. Adebamowo, Sally N. Adebamowo & Omolara Oluwasola-Taiwo - and AADM Study Coordinators - Susan Nkem and Adela Ogundeji. This research was supported by the Intramural Research Program of the Center for Research on Genomics and Global Health (CRGGH). The CRGGH is supported by the National Human Genome Research Institute, the National Institute of Diabetes and Digestive and Kidney Diseases, the Center for Information Technology, and the Office of the Director at the National Institutes of Health (1ZIAHG200362).

## Author contributions

A.A.A. and C.N.R. conceptualized and supervised the project. A.A.A. and A.P.D. designed the study. E.F.H. and A.P.D. managed tissue collection, planned experiments, and drafted the manuscript. E.F.H., S.A., M.R.K., and S.W. isolated and imaged samples. F.N., S.I., and S.D. performed and supervised sequencing. E.F.H. performed single-cell analysis. All authors reviewed and approved the manuscript.

## Funding

## Competing interests

The authors declare no competing interests.

## Ethics

Ethical approval for the AADM study under which the samples were collected was obtained from the National Institutes of Health (09-HG-N070) and the National Health Research Ethics Committee of Nigeria (NHREC). Informed consent was obtained from all participants prior to enrollment. The research adhered to the Declaration of Helsinki.
