## [Transparent Peer Review file · Communications Biology]

Optimized Methods for scRNA-seq and snRNA-seq of Skeletal Muscle Stored in Nucleic Acid Stabilizing Preservative

Corresponding Author: Dr Adebawale Adeyemo

Version 0:

Reviewer comments:

Reviewer #1

(Remarks to the Author)

Dear Authors,

Your study demonstrates that archived human skeletal muscle tissue stored in Allprotect Tissue Reagent (ATR) can produce high-quality data suitable for single-cell and single-nucleus RNA sequencing. The study tested four protocols—filtered cells, filtered nuclei, sorted cells, and sorted nuclei—and found that all methods generated biologically relevant profiles reflecting the heterogeneity of skeletal muscle. Notably, the study revealed that while sorting samples improved the quality by removing debris and red blood cells, it reduced the overall number of captured samples. However, when tissue quantity is limited, avoiding sorting may maximize sample capture.

What I find also very important novelty of this study is that ATR as a preservative reagent was tested for usage in single cell technology for the first time. The demonstration of suitability for this preserving agent for single cell downstream experiments is very valuable.

Overall, the protocol offers a valuable resource for expanding the applicability of single-cell sequencing to archived clinical and research tissue biospecimens, facilitating more diverse and collaborative research efforts.

The study is nicely planned and conducted. The method section is extensive, providing information needed for the reproduction of the experiments.

I only have a minor suggestion:

Figure 2b – the numbers on the y-axis of violin plots are so small that are difficult to read. This also applies to other violin plots as well. Please correct so the numbers are bigger.

Reviewer #2

(Remarks to the Author)

This paper looks at the feasibility of storage of skeletal muscle samples for downstream sequencing applications in ATR (Allprotect Tissue Reagent). The authors then use 4 tissue pathways and indicate that the 2 samples that are processed generate high quality sc-RNA and sn-RNA sequencing data. This is useful for remote sites but several points should be clarified to increase the impact of this paper:

1. Although there is a protocol for cell isolation and for imaging, there is no protocol guiding the collection, storage or transportation of these samples. It would be helpful for the authors to comment on: the quantity of material collected in the biopsy, the immediate processing of the samples (it is mentioned that the samples are dissected but not whether they are washed, the quantities of samples which are stored per aliquot, the type of tube utilised, the temperature at which the samples are stored and transported etc. These should be considered since the authors are claiming that this may represent an attractive solution to allow for RNA sequencing in samples from LMIC.
2. Can the authors explain why skeletal muscle was chosen as a source material? Was this a convenience sample or was it chosen because of the skeletal specific gene expression. Could the authors speculate as to whether this solution could be extrapolated to tissue with high endogenous levels of RNAses (like the pancreas, for example)?
3. Could the authors provide any limitations to this methodology. Although the expression profiles seem to support the cell types were present as expected, could the authors also suggest where they would consider the different methodologies.

Version 1:

Reviewer comments:

Reviewer #1

(Remarks to the Author)

Thank you for improving the figures. I have no further comments.

Reviewer #2

(Remarks to the Author)

I am happy that my concerns have been adequately addressed.

Response to reviewers' comments

COMMSBIO-24-3889-T: "Optimized Methods for Obtaining scRNA-seq and snRNA-seq Data from Skeletal Muscle Stored in Nucleic Acid Stabilizing Preservative"

Reviewers' comments:

Reviewer #1 (Remarks to the Author):

Dear Authors,

Your study demonstrates that archived human skeletal muscle tissue stored in Allprotect Tissue Reagent (ATR) can produce high-quality data suitable for single-cell and single-nucleus RNA sequencing. The study tested four protocols—filtered cells, filtered nuclei, sorted cells, and sorted nuclei—and found that all methods generated biologically relevant profiles reflecting the heterogeneity of skeletal muscle. Notably, the study revealed that while sorting samples improved the quality by removing debris and red blood cells, it reduced the overall number of captured samples. However, when tissue quantity is limited, avoiding sorting may maximize sample capture.

What I find also very important novelty of this study is that ATR as a preservative reagent was tested for usage in single cell technology for the first time. The demonstration of suitability for this preserving agent for single cell downstream experiments is very valuable.

Overall, the protocol offers a valuable resource for expanding the applicability of single-cell sequencing to archived clinical and research tissue biospecimens, facilitating more diverse and collaborative research efforts.

The study is nicely planned and conducted. The method section is extensive, providing information needed for the reproduction of the experiments.

The authors are grateful for the feedback.

I only have a minor suggestion:

Figure 2b – the numbers on the y-axis of violin plots are so small that are difficult to read. This also applies to other violin plots as well. Please correct so the numbers are bigger.

We appreciate the reviewer noting this issue. We have now increased the font size of the numbers on the y-axis of the plots in Figures 2 and 4 as well as in the Extended Data.

Reviewer #2 (Remarks to the Author):

This paper looks at the feasibility of storage of skeletal muscle samples for downstream sequencing applications in ATR (Allprotect Tissue Reagent). The authors then use 4 tissue pathways and indicate that the 2 samples that are processed generate high quality sc-RNA and sn-RNA sequencing data. This is useful for remote sites but several points should be clarified to increase the impact of this paper:

The authors appreciate the summary of this work.

1. Although there is a protocol for cell isolation and for imaging, there is no protocol guiding the collection, storage or transportation of these samples. It would be helpful for the authors to comment on: the quantity of material collected in the biopsy, the immediate processing of the samples (it is mentioned that the samples are dissected but not whether they are washed, the quantities of samples which are stored per aliquot, the type of tube utilised, the temperature at which the samples are stored and transported etc. These should be considered since the authors are claiming that this may represent an attractive solution to allow for RNA sequencing in samples from LMIC.

The authors appreciate the reviewer's point that the collection and storage of samples is a relevant part of this protocol. To address these comments, we have included the following details starting at line 238 in the revised manuscript:

Skeletal muscle samples were obtained from the vastus lateralis muscle under local anesthesia using the Vacora™ Vacuum Assisted Biopsy System, a handheld biopsy system capable of collecting up to 170 mg of tissue in one pass. The vastus lateralis muscle was used as the biopsy site because of its bulk, distance from major vascular structures, and easy accessibility. The resulting samples were quickly dissected into visible skeletal muscle and adipose tissue components, and each component was immediately placed in a 15-ml prelabelled Falcon centrifuge tube and pre-filled with at least 5 ml of Allprotect® Tissue Reagent (ATR) (Qiagen, Germantown, MD, catalogue no. 76405) to stabilize tissue RNA, DNA and proteins. The AllProtect volume was accordingly adjusted to the size of the tissue aliquot which in our study varies between 50 mg to 180 mg to ensure complete submersion of the tissue samples in the collection tubes. The collection tubes were then placed in an ice bucket and transfer from the clinic to the lab in couple of hours. For archival of the tissue samples at -80 °C, they are kept overnight in the solution at 4-8 °C and then transferred to -80 °C until shipped to the coordinating center, the Center for Research on Genomics and Global Health at the National Institutes of Health (Bethesda, MD, USA) from the study site (Ibadan, Nigeria). Shipping of the samples was done on dry ice.

We have updated the manuscript methods and the online methods to reflect these additions.

2. Can the authors explain why skeletal muscle was chosen as a source material? Was this a convenience sample or was it chosen because of the skeletal specific gene expression. Could the authors speculate as to whether this solution could be extrapolated to tissue with high endogenous levels of RNAses (like the pancreas, for example)?

Skeletal muscle was specifically chosen as the source material because the samples were collected as part of a type 2 diabetes study (the Africa America Diabetes Mellitus (AADM) study) and skeletal muscle is one of the key tissues involved in the pathophysiology of type 2 diabetes.

Thus far, we have only tested the protocol on human muscle samples. We hesitate to speculate about if the solution could be extrapolated to other tissues given that, in addition to endogenous levels of RNAses, other factors such as tissue composition, tissue density and inter-species differences may affect the yield of the protocol. However, we note that Qiagen, the manufacturer of All Protect Tissue Reagent, has a poster detailing RNA stabilization of multiple *rat* tissues (i.e., intestine, heart, lung, spleen, uterus, brain, liver, bone, stomach). While it is likely that All Protect Tissue

Reagent will work in these rat tissues as long as the manufacturer's tissue-to-reagent ratios are maintained, empirical tests of *human* tissues are needed. This is addressed at line 214.

3. Could the authors provide any limitations to this methodology. Although the expression profiles seem to support the cell types were present as expected, could the authors also suggest where they would consider the different methodologies.

Although this method works exceptionally well on solid tissue, one limitation is that it may not work as well on cells in suspension (i.e., blood or homogenized tissue), without significant changes to the protocol. Manufacturers of All Protect Tissue Reagent also note that the preservative can cause cell shrinkage, making it inappropriate for morphological studies. Finally, proteins are denatured and suitable for western blotting but not for structural assays.

With regards to our protocol for extraction and isolation of single cells and nuclei for RNA sequencing, we have not tested recovery of intact chromatin or preservation of RNA/RNA, RNA/DNA, or DNA/DNA interactions. Similarly, care must be taken when selecting antibodies for flow cytometry-based enrichment as protein denaturation could compromise conformational epitopes. These cautions have been added to the *Discussion* section of the manuscript at line 213.